# A street-view-based method to detect urban growth and decline: A case study of Midtown in Detroit, Michigan, USA

**Giyoung Byun**, **Youngchul Kim***

KAIST Urban Design Lab, Department of Civil and Environmental Engineering, Korea Advanced Institute of Science and Technology, Daejeon, Korea

* youngchulkim@kaist.ac.kr

**Data Availability Statement:** The data and codes that support the findings of this study are available in figshare.com with the identifier https://figshare.com/s/cb3f6bd72133fefcfb8c.

## Abstract

Urban growth and decline occur every year and show changes in urban areas. Although various approaches to detect urban changes have been developed, they mainly use large-scale satellite imagery and socioeconomic factors in urban areas, which provides an over-view of urban changes. However, since people explore places and notice changes daily at the street level, it would be useful to develop a method to identify urban changes at the street level and demonstrate whether urban growth or decline occurs there. Thus, this study seeks to use street-level panoramic images from Google Street View to identify urban changes and to develop a new way to evaluate the growth and decline of an urban area. After collecting Google Street View images year by year, we trained and developed a deep-learning model of an object detection process using the open-source software TensorFlow. By scoring objects and changes detected on a street from year to year, a map of urban growth and decline was generated for Midtown in Detroit, Michigan, USA. By comparing socioeconomic changes and the situations of objects and changes in Midtown, the proposed method is shown to be helpful for analyzing urban growth and decline by using year-by-year street view images.

## 1 Introduction

Urban growth and decline are processes that show changes in urban areas. While urban growth is a process to newly create various urban elements [1], urban decline means that the entire or part of an urban area becomes devastated and abandoned. For example, urban growth includes spatial extension, population increase, and economic growth [2, 3], and urban decline includes empty houses, abandoned buildings, and job losses [3, 4]. Since urban decline is an undesirable condition in cities, various policies are used to try to change and protect urban areas from urban decline. When analyzing the situation of urban growth and decline, it is possible to prepare for urban planning and management to sustain daily places in urban areas. Thus, as urban growth and decline have occurred, relevant studies have examined and proposed models to analyze urban growth and decline.

**Funding:** This research was supported by Urban Declining Area Regenerative Capacity-Enhancing Technology Research Program (22TSRD-C151228-04) and 'Innovative Talent Education Program for Smart City' from the Korea Agency for Infrastructure Advancement funded by Ministry of Land, Infrastructure and Transport of Korean government and the EDISON Program through the National Research Foundation of Korea (NRF) funded by the Ministry of Science & ICT (NRF-2017M3C1A6075020). The funders had no role in study design, data collection and analysis, decision to publish, or preparation of the manuscript.

**Competing interests:** The authors have declared that no competing interests exist.

To identify whether urban growth or decline occurs, previous studies have used socioeconomic aspects such as real estate market conditions [3–6], chronological economic status [7], real incomes [8], and housing conditions [9]. Those studies demonstrated theoretical and empirical evidence to understand urban growth and decline. Using spatial and statistical data, an urban growth model was proposed to include commonly available, spatially specific data points [10]. Those previous studies have a strength to analyze and apply for every region and country using spatial and statistical data. Although those relevant studies effectively explained the characteristics of urban growth and decline, they normally demonstrate overall changes and rarely explain street-level changes where people explore for their daily life. Since people explore places and notice changes daily at the street level, it would be useful to develop a method to identify urban changes at the street level and demonstrate whether urban growth or decline occurs there. Street-level analysis helps develop local-oriented specific strategies focusing on where people daily explore and easily identify changes for the region by appropriate solutions. Therefore, this study seeks to use street-level views to identify urban changes and to develop a new way to evaluate the growth and decline of an urban area. In particular, yearly changes in street view images motivate this study to focus on urban changes at a street level. This novel method to analyze urban changes at a street level seeks to contribute to automation and efficiency in identifying growth and declines in urban areas.

## 2 Computational approaches to analyze urban growth and decline

As computational technology improves, various studies have sought to adopt computational approaches to analyze urban growth and decline. Maithani [11] studied built-up maps and nonbuilt-up maps and used them to train an artificial neural network (ANN). To simulate urban growth in Saharanpur, India, Maithani [11] evaluated the Kappa index and three specific spatial metrics. Park et al. [12] studied urban growth using a land suitability index and mapping with a geographic information system (GIS). They focused on predictions and comparisons among the indexes. They used the frequency ratio (FR), an analytical hierarchy process (AHP), logistic regression (LR), and an ANN to forecast urban land-use changes. Mohammady et al. [1] used an ANN to create a tool to simulate urban growth in Sanandaj, Iran. This study is important because it used satellite images to predict urban growth with a neural network (NN). Tayyebi et al. [13] also used an ANN to predict urban growth patterns and compared the outcomes with an urban growth boundary model. Guan et al. [14] used other computational methods to analyze urban growth; specifically, they used a cellular automata model on an ANN to simulate and forecast urban growth. The data were land-use geospatial data from Beijing, China, and were analyzed by cell. Alkheder and Shan [15] used remote sensing imagery and an NN algorithm to simulate urban growth boundaries. They posited seven classes of urban components: water, roads, residential areas, commercial areas, forests, pasture grasses, and row crops. They used two algorithms, the simple adaptive linear neural network (SALNN) and a backpropagation neural network algorithm (BPNN), and the experimental data were six types of satellite images of Indianapolis, Indiana, USA. This study was valuable in that it combined an NN with urban growth. Additionally, the research demonstrated the potential of implementing NN algorithms as a tool to predict urban growth patterns based on historical satellite images.

Deep convolutional neural networks have become a significant approach to analyze urban growth and decline. Vakalopoulou et al. [16] proposed a building detection model with an automated system using a deep convolutional neural network (CNN) with high-resolution remote sensing data. They used satellite images of Greece acquired between 2006 and 2011. The contribution of this study is the development of several methods of urban building

detection. However, if these types of deep learning algorithms were used to detect urban changes, a major contribution could be made to the research on urban changes. Zhou and Gong [17] used LiDAR data to detect buildings in urban areas. They acquired airborne LiDAR data, converted it into grayscale images, and applied a CNN to detect buildings in the grayscale images. This method requires less preprocessing of the LiDAR data and has the possibility of being applied to disaster recovery in urban areas. NNs have also been applied to studies of specific urban components. Fang et al. [18] used Faster R-CNN to detect workers and heavy equipment at construction sites. They applied an NN model to urban construction sites. This study contributes by identifying people and equipment in unsafe areas. Additionally, it could lead to major improvements in safety at construction sites.

# 3 Material and methods

This study seeks to develop a computational model to analyze urban growth and decline by determining street-level changes from year to year. To automatically measure urban growth and decline in an urban area, this study uses street views that have been archived in an open database, Google Street View (GSV). To analyze urban changes, urban scenes and views must be collected. Since urban scenes and views have become valuable information to provide people for navigating places on the internet, some companies have archived street views. For example, Google provides a map service for those who are interested in collecting location information and navigating specific areas on the internet. Accordingly, Google Maps provides an application programming interface (API) of GSV that helps people use the street view images provided by Google. GSV images have become popular for analyzing urban scenes and perceptions, such as street greenery [19, 20], street score and perception [21–23], and transportation influences [24, 25]. Thus, this study follows three steps: collecting data, applying an algorithm, and identifying changes in urban growth and decline in an urban area. In the data collection step, we collected GSV image data from Google Maps. We chose Detroit, Michigan, USA, for the experiment and collected 360-degree panoramic images that captured the streets of the Midtown area in Detroit. After collecting the images, we applied the data to the TensorFlow object detection API. We used the TensorFlow object detection API to detect factors on streets that can be used to evaluate urban growth and decline at the street level. The TensorFlow object detection API uses a CNN to detect specific elements of the street view images in the experiment. We used Faster R-CNN, a pretrained algorithm, and newly trained our images to detect additional factors. After the detection step, we evaluated urban areas by scoring changes on streets.

## 3.1 Data collection

**3.1.1 Study area: Midtown in Detroit, Michigan, USA.** Midtown in Detroit, Michigan, USA, was selected as the study area. Midtown is the center of Detroit with various economic, educational, cultural and shopping facilities. Midtown has an area of 2.09 square miles with two zip codes, 48201 and 48202, and its population was approximately 14,550 in 2010, which is approximately 2% of Detroit's population. Fig 1 shows Midtown.

Data on buildings, blocks, and streets were collected from OpenStreetMap by using the OpenStreetMap API. We used OpenStreetMap Data including buildings, blocks, and streets in December 2018 as a base map to compare. This study focuses on streets and nearby areas where public transportation routes exist. Various daily activities, such as commuting and shopping, normally occur along these public transportation routes. Public transportation routes are an important factor when planning sustainable cities and urban development [26].

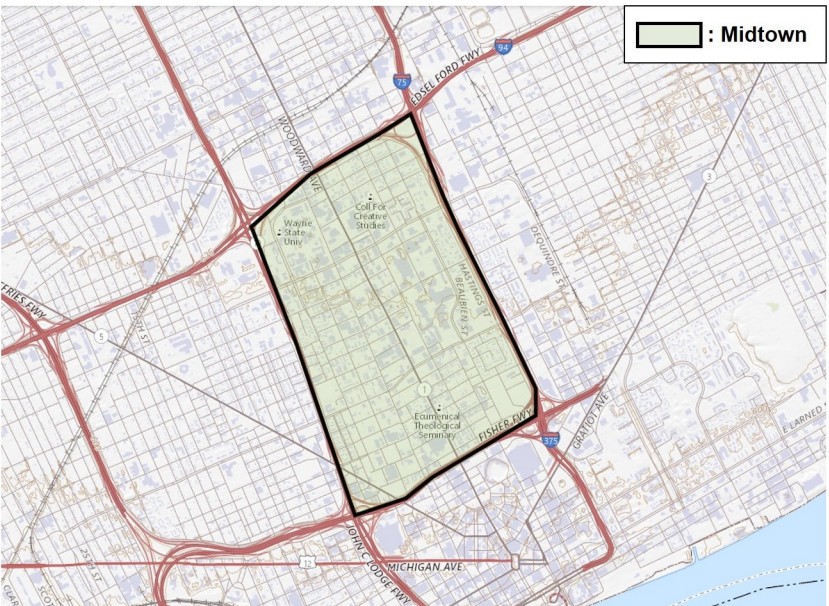

**Fig 1. The view of Midtown in Detroit, Michigan, USA (source: USGS national map viewer).**

Additionally, newly urban areas are developed along major transportation routes [27]. Fig 2 shows the selected transportation routes on a map of Midtown.

**3.1.2 Street view data.** GSV images of Midtown were collected. GSV on Google Maps is a virtual representation of scenes on streets using millions of panoramic images. GSV supports time stamps for each location. For example, each location in Midtown has various time stamps from 2007 to 2018. GSV images have their own parameters at web addresses. The parameters consist of location, panorama, size, key, signature, heading, field of view (FOV), pitch, radius and source. The parameters of location, panorama, size, and key are required to collect a panoramic image from GSV, and the parameters of signature, heading, FOV, pitch, radius and source are optimal to collect a panoramic image. According to Street View Developer Guide Documentation (https://developers.google.com/maps/documentation/streetview/overview), we collected panoramic images from GSV by submitting the appropriate parameters. With the GSV API, panoramic images of Midtown were collected for 2009, 2011, 2013, 2015 and 2017. To collect GSV images, we used Python 3.6 to obtain panoramic views from GSV by time period in Midtown. Following the public transportation routes in Fig 2, we chose to score objects on 17 urban streets, which are listed in Table 1. GSV images were collected for the 17 chosen streets. In total, 7427 images were collected.

## 3.2 Algorithm development of a CNN with a TensorFlow object detection API

**3.2.1 TensorFlow object detection API, CNN, Faster R-CNN.** This study uses the Google TensorFlow object detection API. TensorFlow developed by Google is an open source software tool library that uses a machine learning engine (found at https://github.com/tensorflow/models/tree/master/research/object_detection). In particular, the TensorFlow object detection API that this study uses is easy to install, and train and organizing object detection is straightforward in the API. By using the TensorFlow object detection API model, we choose a CNN model that is specified for image recognition. Since the CNN detects objects in images or

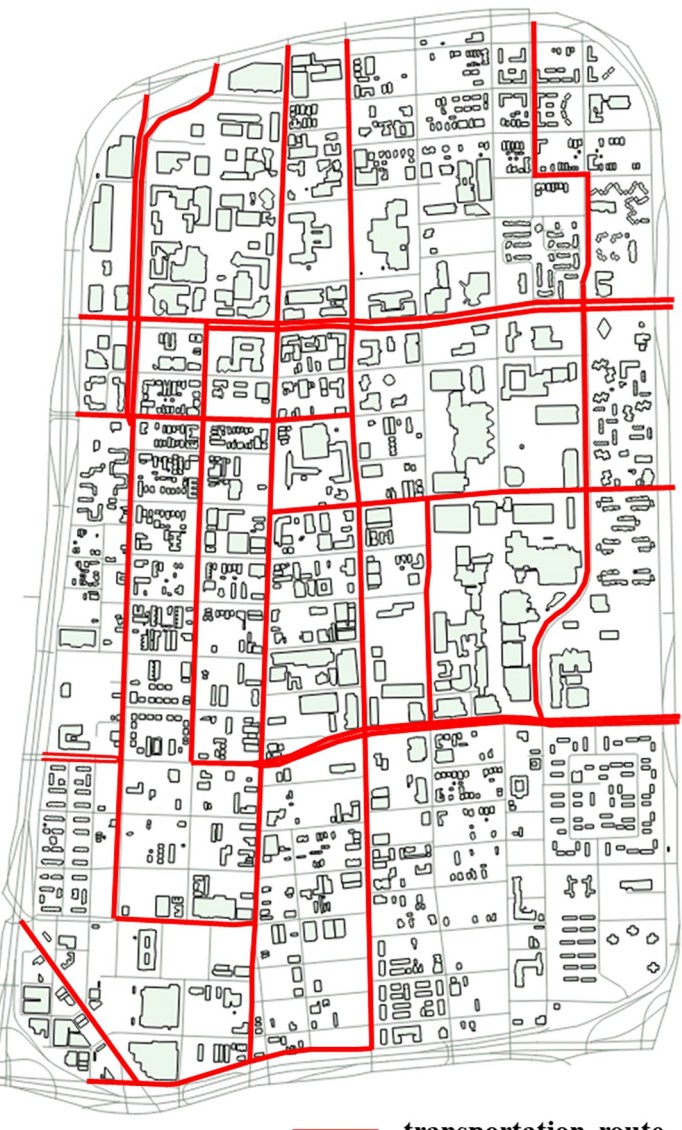

transportation route

**Fig 2. Public transportation routes in Midtown.**

videos, extracts features from images and uses the images for processing, it offers high resolution during image processing. In particular, this paper uses a specific Faster R-CNN model, the faster_rcnn_resnet101_coco model of the TensorFlow object detection API. Using this algorithm, this study follows the experimental steps in Fig 3, which shows the training and detection processes.

**3.2.2 Using the TensorFlow object detection API and training our own data.** Using this TensorFlow object detection API, we selected the faster_rcnn_resnet101_coco model (found at https://github.com/tensorflow/models/blob/master/research/object_detection/g3doc/detection_model_zoomd). It was necessary to use our data, not pretrained data, meaning that there were additional processes in the experiment. First, we downloaded the TensorFlow object detection API. We also selected the faster_rcnn_resnet101_coco model and copied it to the TensorFlow object detection API folder. Second, we collected testing and training data for

**Table 1. Numbers of images collected for the selected streets.**

| Street name | Year | | | | |
|---|---|---|---|---|---|
| | **2009** | **2011** | **2013** | **2015** | **2017** |
| **Woodward Avenue** | 95 | 141 | 72 | 274 | 215 |
| **2nd Avenue** | 123 | 103 | 55 | 88 | 98 |
| **3rd Avenue (west side)** | 155 | 149 | 130 | 167 | 202 |
| **3rd Avenue (east side)** | 81 | 69 | 59 | 73 | 90 |
| **Beaubien Street** | 61 | 51 | 49 | 41 | 32 |
| **Canfield Street** | 81 | 64 | 34 | 62 | 100 |
| **Cass Avenue** | 194 | 185 | 130 | 189 | 246 |
| **Grand River Avenue** | 33 | 31 | 39 | 23 | 32 |
| **John R Street** | 50 | 37 | 24 | 39 | 57 |
| **Mack Ave (west side)** | 106 | 132 | 95 | 117 | 60 |
| **Mack Ave (east side)** | 107 | 94 | 97 | 88 | 113 |
| **St Antoine Street** | 143 | 114 | 113 | 109 | 106 |
| **Temple Street** | 34 | 28 | 11 | 21 | 11 |
| **Warren Avenue (west side)** | 109 | 89 | 56 | 82 | 77 |
| **Warren Avenue (east side)** | 128 | 80 | 80 | 95 | 99 |
| **W Fisher Avenue** | 61 | 59 | 43 | 38 | 26 |
| **W Forest Avenue** | 66 | 69 | 47 | 42 | 59 |
| **Total** | 1627 | 1495 | 1134 | 1548 | 1623 |

the experiment. We downloaded all of the images of the selected factors that can affect urban growth and decline. After downloading the images, we manually labeled images for training using a labeling program. Third, after the labeling process, we trained our factors on the TensorFlow object detection API model. We experimentally trained data at a rate of 80% and tested data at a rate of 20% for each factor. Labeled images have different image sizes, and an

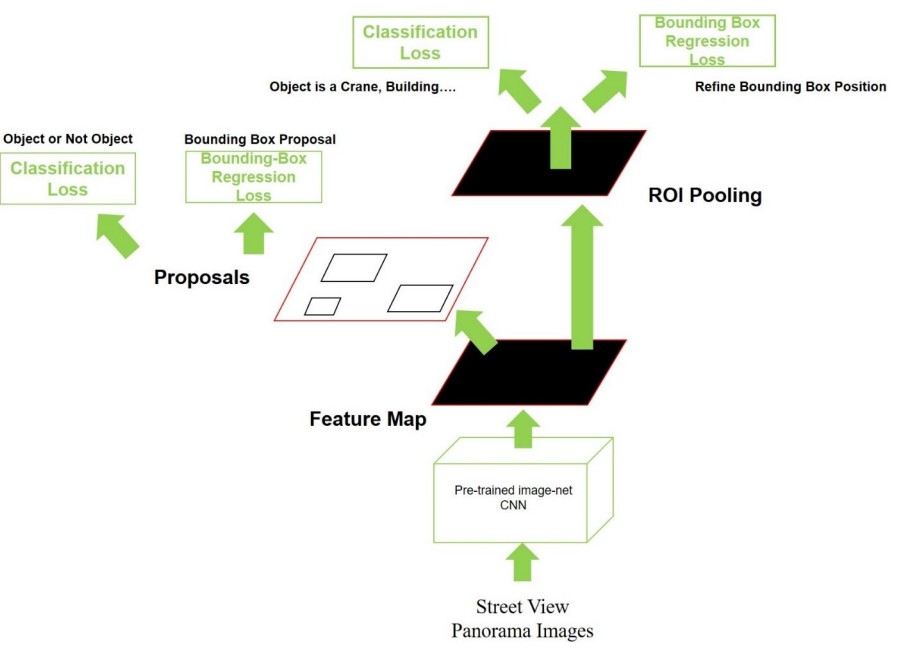

**Fig 3. Experimental steps used in this study.**

additional step should be performed when creating the test images for the experiment. We used a sample GSV panoramic image size of 1664 by 832.

Next, we describe the steps by which we collected the numerous images that were used for training. The TensorFlow object detection API requires one hundred or more images of an object to train a good detection classifier. Before we trained and tested our factor image data with the GSV panoramic images, it was necessary to gather all of the images to set the factors. We used Python web crawling when collecting the training data. After gathering all of the images, it was necessary to label each image for training. We used LabelImg, a free open source program (found at https://github.com/tzutalin/labelImg). When we used the TensorFlow object detection API, a data preprocessing step was necessary. LabelImg creates XML files, which are used to train images with the TensorFlow model. LabelImg is a graphical image annotation tool written in Python. The graphical interface was made with PyQt. Annotations can be saved as XML files in the PASCAL VOC format, which is used by ImageNet. In addition, ImageNet supports the YOLO format. With these labeled images, we made TFRecords that serve as input data to the TensorFlow training model. We used a Python program to convert the XML files to the CSV file format. We also generated a TFRecord Python script called Dat Tran's Raccoon Detector dataset (found at https://github.com/datitran/raccoon_dataset). During this process, it was necessary to change the row label values to match our trained object. Next, the original label map file of the TensorFlow object detection API model was adjusted. We created a label map and edited the training configuration file. The label map defines the mapping class names to class ID numbers to tell the trainer what each object is. We used a text editor to create a new file for the experiment and saved it as a labelmap.pbtxt file in the TensorFlow object detection folder. Finally, we trained the images using the TensorFlow object detection API model. The accuracy indicator of the deep learning models we trained was approximately 86%, and examples of detected results for each element are shown in Figs 4–7.

### 3.3 Detecting urban growth and decline

**3.3.1 Growth and decline factors.** To determine whether an urban area grows or declines, we need to decide which factors are significant for detecting urban changes. Then, an urban area can be evaluated based on the detected factors. It is important to create scores using detected factors from GSV images. Streets are scored according to the detection of objects that relate to urban growth and decline. After detection in the TensorFlow object detection API, we create growth and decline factors that indicate urban changes. We selected factors that have been demonstrated to affect changes in cities by previous studies, including factors

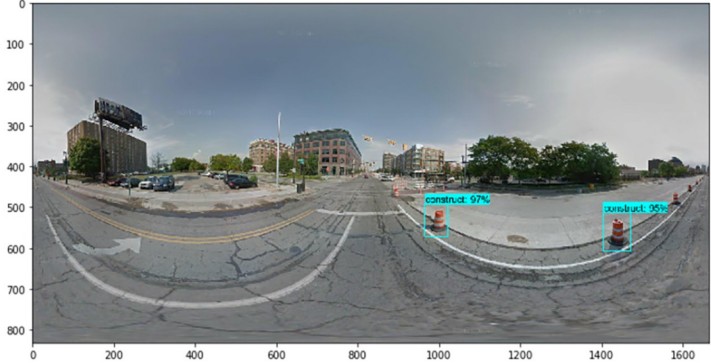

**Fig 4. Detected results of traffic barrels (i.e., construction).**

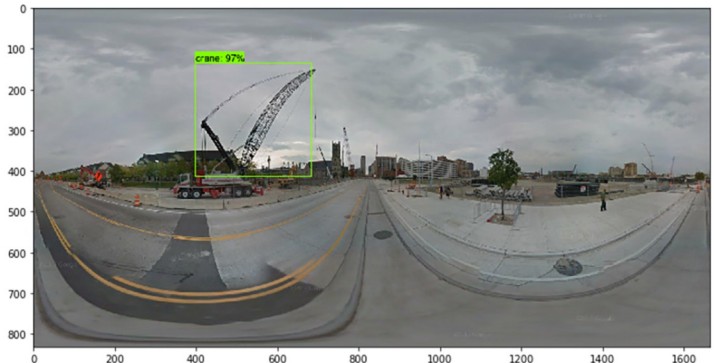

**Fig 5. Detected result of a crane.**

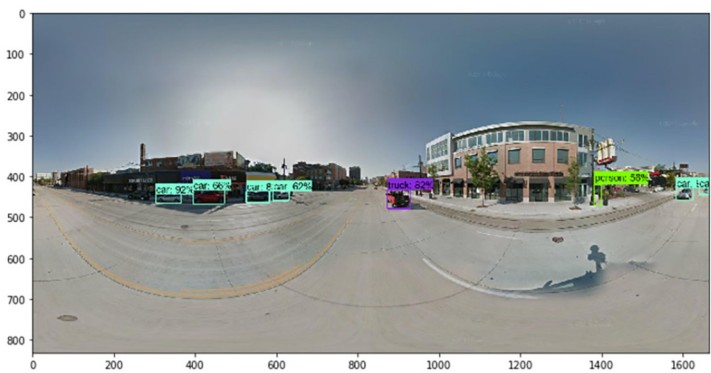

**Fig 6. Detected results of a person and cars.**

related to building construction [28–32] and transportation elements [33–39]. Each factor has a score ranging from -0.5 to +0.5 to have a single scale. When a street has a positive score close to +0.5, it is in growth or will grow. On the other hand, when a street has a negative score close to -0.5, it is in decline or will decline. We assign scores according to detected factors on the GSV images for two years. If growth factors are newly shown, we add points to the coordinates, whereas if decline factors are detected compared to earlier images, we assign a decline score for each image.

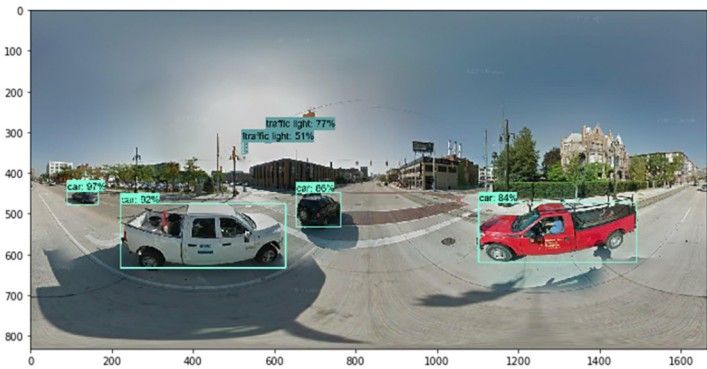

**Fig 7. Detected results of cars and traffic lights.**

**3.3.2 Growth and decline factors in GSV images.** We chose six growth factors to evaluate the growth score and its impacts on the spatial change by considering its size and frequency in collected street view images. The first factor is buildings. If a new building is in an image compared to images from past years, it means that the street has been developed [28]. The more buildings are constructed, the greater the population density. As the number of buildings increases, urbanization and urban growth are shown [29]. Thus, we score +0.5 for new buildings. Additionally, the new construction of buildings can be a source of economic capacity or can increase economic capacity [30]. The second factor is a crane, which indicates a construction site. It is a main factor to construction site. A crane should inevitably be used when a building is constructed [31]. When a crane is captured, we score +0.3 for that image. The third factor is a traffic barrel, which is used to signify a construction site. Traffic barrels are a common sign that road work or other types of construction take place near or on a road [32]. If a street has a construction site and traffic barrels, we score +0.03 for each barrel, showing that urban growth is in the construction process. Constructing new buildings and housing is a significant factor that improves the economic conditions in a city [33]. The fourth factor is a bus stop. If we detect new bus stops, we score them +0.2 points each. Bus stops and traffic lights are parts of transportation systems. Accessibility to bus stops is usually used to measure urban growth and can be an indicator of urban sprawl [34, 35]. Since changes in urban transportation can affect urban employment [36], new components of transportation change public space. For the fifth factor, if there is a new traffic light, we score +0.1 points for urban growth. Traffic Lights show the traffic volume of vehicles [37]. They can be urbanization infrastructures as traffic management systems [38]. The sixth factor includes people, cars, and bicycles, which can affect urban activation and traffic flow. For street activation, walkways and bikeways are necessary [39]. If the traffic flow of the region is concentrated, infrastructural facilities are located at the center of the town [40]. Scores of people, cars, and bicycles are given +0.02 points each.

Regarding the decline factors on a street, we chose the loss of similar factors. The first factor is the loss of a building. The loss of a building has a huge impact on a street. Although the loss of a building can be explained in many ways, deconstruction obviously occurs at the location. Since deconstruction can affect the deactivation of a street [41], we score the loss of a building at -0.5 points. For the second decline factor, if there is no person in the view, we score this situation at -0.02 points each. We easily observe many people on an active street. People go shopping or walking on a street if the street is alive. For the third decline factor, if no cars are detected in the view, we score such scenes at -0.02 points each. The disappearance of the growth factors, crane and traffic barrel would mean the end of the construction, so we excluded those factors. Table 2 shows the growth and decline factors used in this study. By detecting these factors on urban streets, we calculated urban growth and decline scores.

**3.3.3 Detecting urban growth and decline factors on streets.** Using the TensorFlow object detection API, we train and test the detection model to detect the factors of urban growth and decline. By collecting relevant images of factors, including traffic barrels, buildings, cranes, and bus stops, we train the model to detect traffic barrels, buildings, cranes, people, and cars. After detecting factors in GSV images, we assign growth and decline points. For example, Figs 4–7 show the results detected by the TensorFlow object detection API. Each element on a street is used to determine the score for that street.

## 3.4 Mapping data of urban growth and decline with a GIS

To visualize the urban growth and decline analysis, we create maps using QGIS. Because the scoring is based on panoramic street views and the captured points of street view images each year are different, we determined that the best way to score growth and decline is to express

**Table 2. Growth and decline factors used in this study.**

| Status | Factor | Score | Description |
|---|---|---|---|
| **Growth** | Building | +0.5 | Potential of new life for people [27, 28, 29, 32] |
| | Crane | +0.3 | Potential of new construction [30, 32] |
| | Traffic barrel | +0.03 | Potential of new construction [31, 32] |
| | Bus stop | +0.2 | Potential of new transportation [33, 34, 35] |
| | Traffic light | +0.1 | Potential of new transportation [36, 37] |
| | Person | +0.02 | Potential of street activation [38] |
| | Car or bicycle | +0.02 | Potential of street activation [38, 39] |
| **Decline** | Loss of building | -0.5 | Potential of street decay [40] |
| | Loss of bus stop | -0.2 | Inconvenience of transportation [33, 34, 35] |
| | Loss of traffic light | -0.1 | Inconvenience of transportation [36, 37] |
| | Loss of person | -0.02 | Deactivation of street [38] |
| | Loss of car or bicycle | -0.02 | Deactivation of street [38, 39] |

the score using a buffer of 200 meters for the locations of the street view images. Fig 8 shows the locations of the street view images in 2017. By using an intersection function in QGIS, we combine the scores with the building data within the buffer. By extracting the mean values of urban street scores, we determine whether an urban street is growing or declining. By analyzing the street view images with the factors detected by the trained TensorFlow object detection API, we calculate the scores of urban growth and decline on streets in Midtown in 2011, 2013, 2015 and 2017.

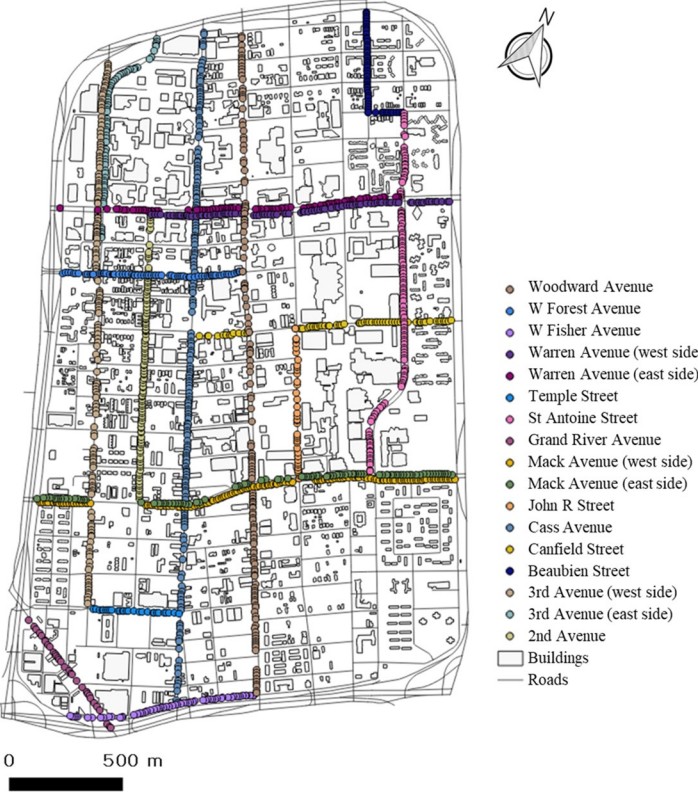

**Fig 8. Locations of street view images in Midtown.**

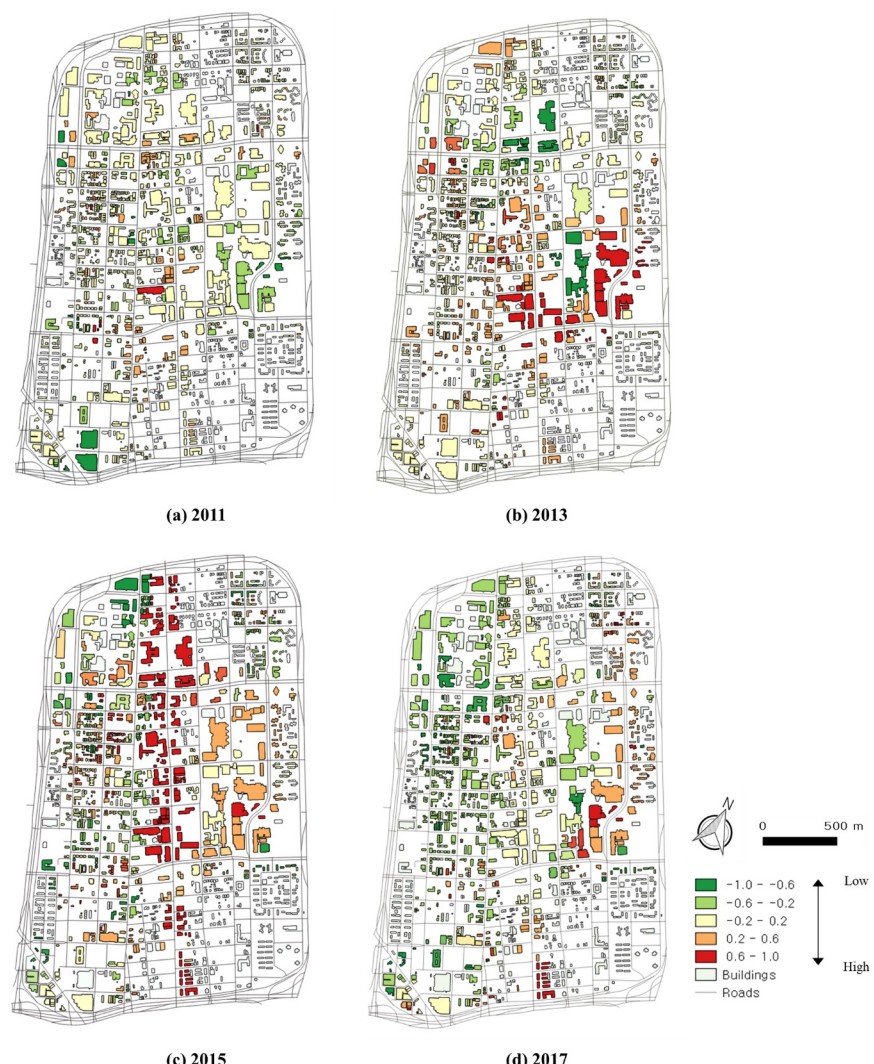

**Fig 9. Urban growth and decline scores in Midtown in (a) 2011, (b) 2013, (c) 2015, and (c) 2017.**

## 4 Results of the urban growth and decline analysis

### 4.1 Urban growth and decline in Midtown in 2011, 2013, 2015 and 2017

According to the scores in Midtown in 2011 in Fig 9(a), significant changes occurred on Woodward Avenue. According to the scores in 2013 in Fig 9(b), high scores are found on Woodward Avenue and St Antoine Street. In 2013, new buildings were constructed on these streets. In 2015, as shown in Fig 9(c), there were significant changes on certain streets. Construction was observed on Woodward Avenue in 2015, and cranes and traffic barrels were detected as construction components. The score for this street in 2015 was higher than that in 2013. Woodward Avenue had a higher score in 2015 than in other years. Because Woodward Avenue was under construction in 2015, many cranes and traffic barrels were detected, more than those in other years. According to Fig 9(a)–9(d), growth situations are detected on Woodward Avenue and other streets.

According to the growth and decline scores, some locations are highlighted in Fig 10 to explain spatial changes. No buildings existed in 2009 or 2011 at the entrance coordinates of

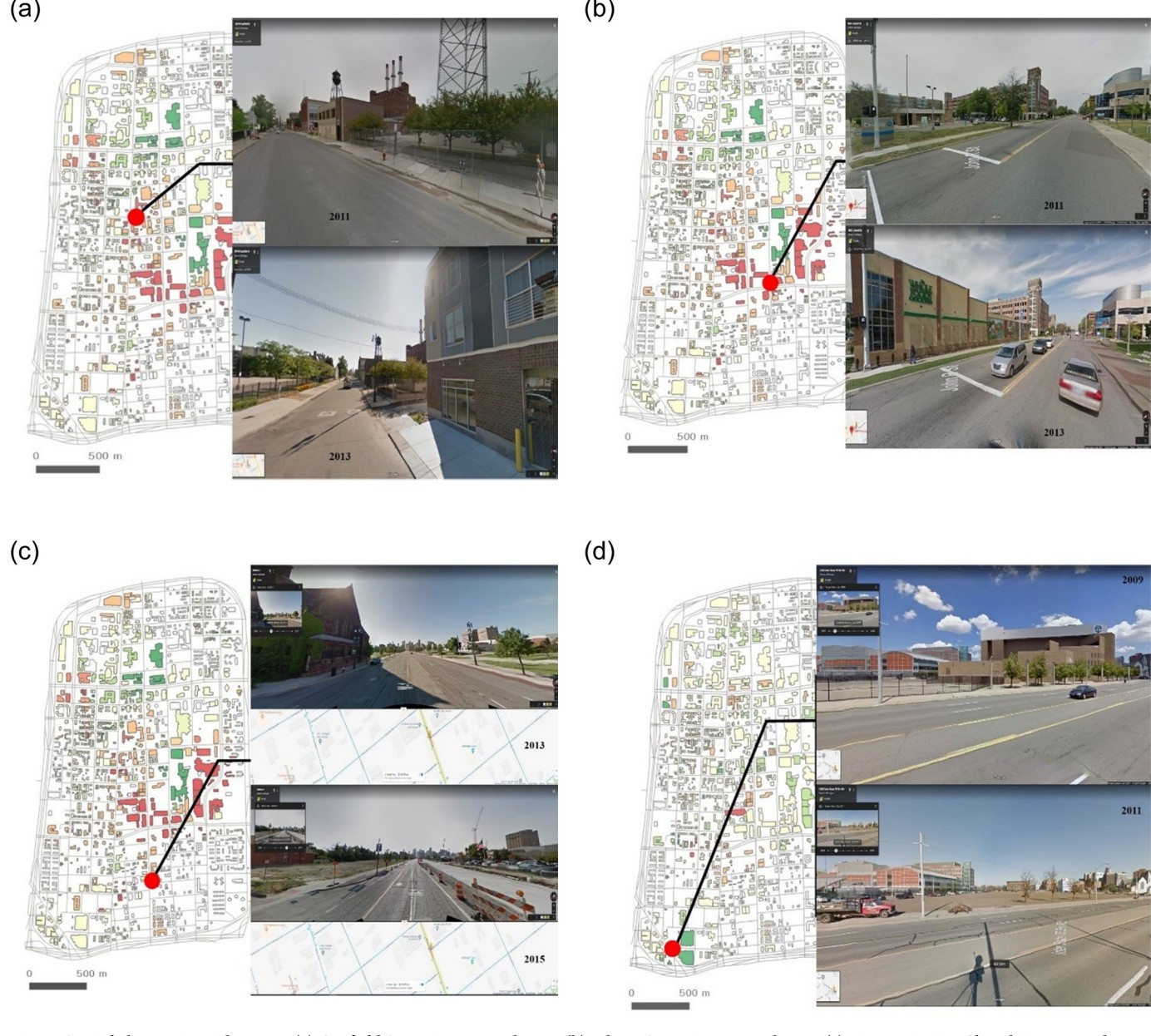

**Fig 10. Spatial changes in Midtown on (a) Canfield Street in 2011 and 2013, (b) John R Street in 2011 and 2013, (c) First Unitarian Church in 2013 and 2015, and (d) Cass Technical High School in 2009 and 2011.**

Canfield Street. However, in 2013, a building emerged, and the street was well maintained, as shown in Fig 10(a). On John R Street, although there were no buildings at the end of the street in 2011, a new building, Whole Foods Market, was built in 2013, as shown in Fig 10(b). Accordingly, the score increased at that location. At the location of the First Unitarian Church of Detroit on Woodward Avenue, although the building was lost, a growth score with cranes and construction evidence was detected in 2015, as shown in Fig 10(c). Cass Technical High School existed at the entrance of Grand River Avenue in 2009 and was lost in 2011, as shown in Fig 10(d). A decline score is detected in 2011 at that location.

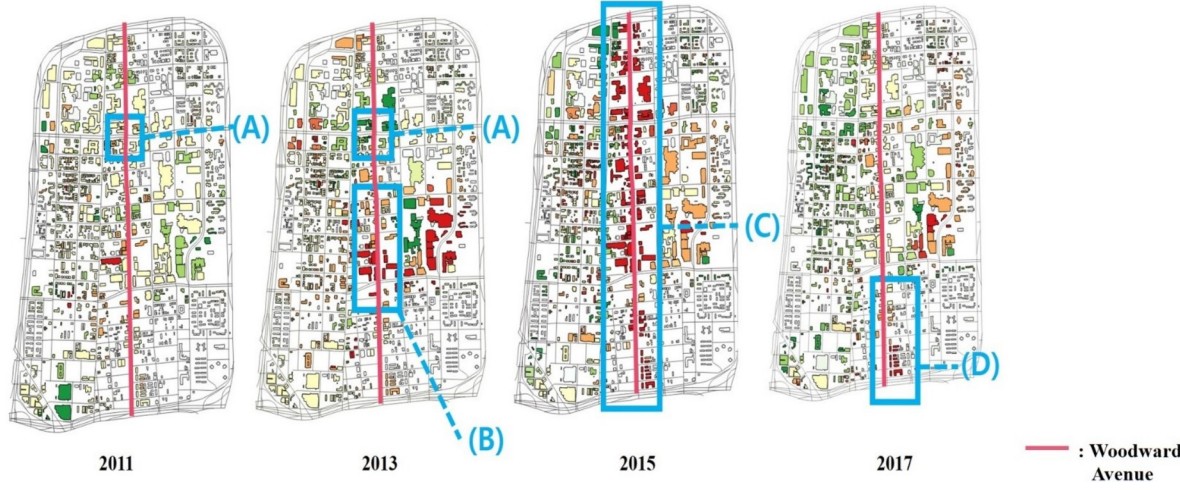

**Fig 11. Changes in Woodward Avenue** —— **(2011, 2013, 2015 and 2017).**

## 4.2 Changes on streets in Midtown

To illustrate the growth and decline changes on each street, a graph of scores is developed to show the mean score of growth and decline at street view locations. After investigating the mean scores of all the streets in Midtown, Woodward Avenue and St Antoine Street are chosen for further discussion of the graphs of their growth and decline scores. Woodward Avenue is a main road that has a long history and important facilities [42]. Additionally, St Antoine Street is a representative area for medical services [43].

Woodward Avenue, mainly located in the center of the district, is a major thoroughfare in Midtown. As shown in Fig 11, there were many changes on the street. One building that stood at location (A) in 2009 was lost by 2011. At the same location (A) in 2013, there were no additional buildings, and the 2011 status was maintained. At location (B), new buildings and construction evidence appeared in 2013. In 2015, nearly the entire street (C) of Woodward Avenue was under construction. The scores for street (C) were higher in 2015 than in 2013. Since the construction was completed in 2015, the score changed less in 2017. In 2017, area (D) of Woodward Avenue was under construction. The score at area (D) increased in 2017. To show these changes, graphs of scores were developed for Woodward Avenue from 2011 to 2017 and are shown in Fig 12. The four panels of Fig 12 show the growth and decline scores on Woodward Avenue in 2011, 2013, 2015 and 2017. According to Fig 12, growth and decline situations are detected on each street. Additionally, the graphs of scores illustrate spatial changes from 2011 to 2017. For example, large growth elements and improvements appeared on many streets in 2015, and Woodward Avenue was the most vibrant among them.

St Antoine Street is located near the Midtown Medical Center. According to Fig 13, the score in area (a) did not significantly change except in the southern area in 2011. However, in the southern area (b) of the street, a new building was constructed in 2013. The score in area (b) increased in 2013. There was continuous new construction in 2015 near the Children's Hospital of Michigan in area (c). The score in area (c) in 2015 thus increased compared to that in 2013 and then increased again in 2017 with the emergence of new buildings in area (d). However, the northern area of St Antoine Street maintained its spatial pattern. According to the graphs of scores on St Antoine Street in Fig 14, while the southern area changed significantly from 2011 to 2017, the northern area maintained its score from 2011 to 2015.

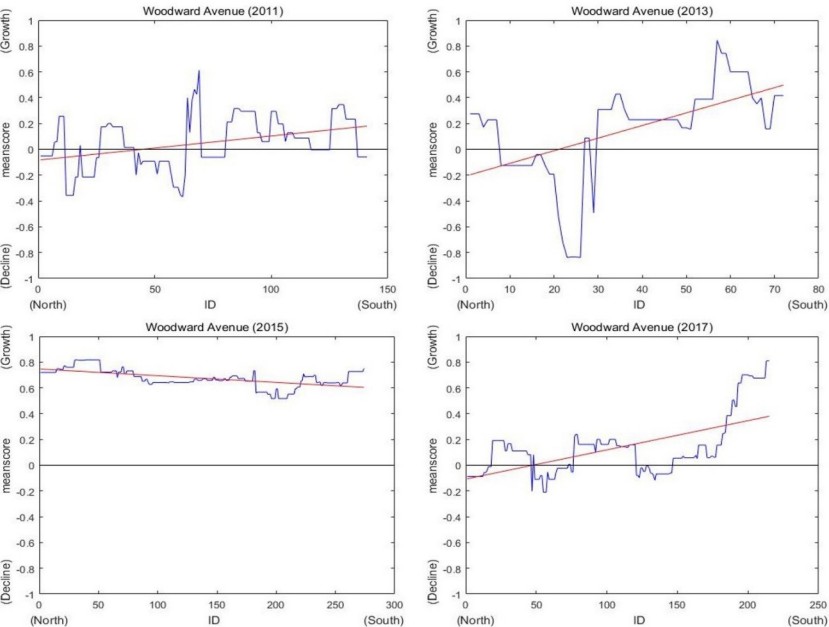

**Fig 12. Graphs of scores on Woodward Avenue (2011, 2013, 2015 and 2017).**

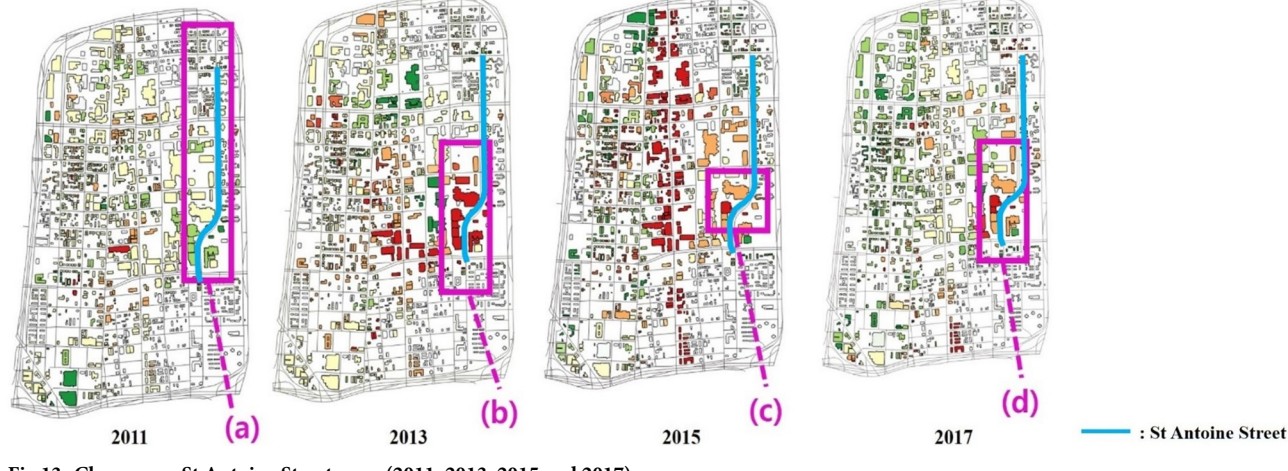

**Fig 13. Changes on St Antoine Street ▬▬ (2011, 2013, 2015 and 2017).**

## 5 Discussion

To verify the results of growth and decline in Midtown, we compare the results of growth and decline with building permits and crime data. A building permit is a type of permission granted by a government or other regulatory agency for the lawful construction of new buildings and extensive work of existing buildings [44]. When any form of work that will modify or add structure to an existing property or land parcel is planned, building permits are necessary [45]. In terms of urban planning, building permits are a necessary part of well-planned urban development. Tasantab [46] used building permits to see how they may be utilized as effective controls on physical development. Building permit data are issued by the Buildings, Safety

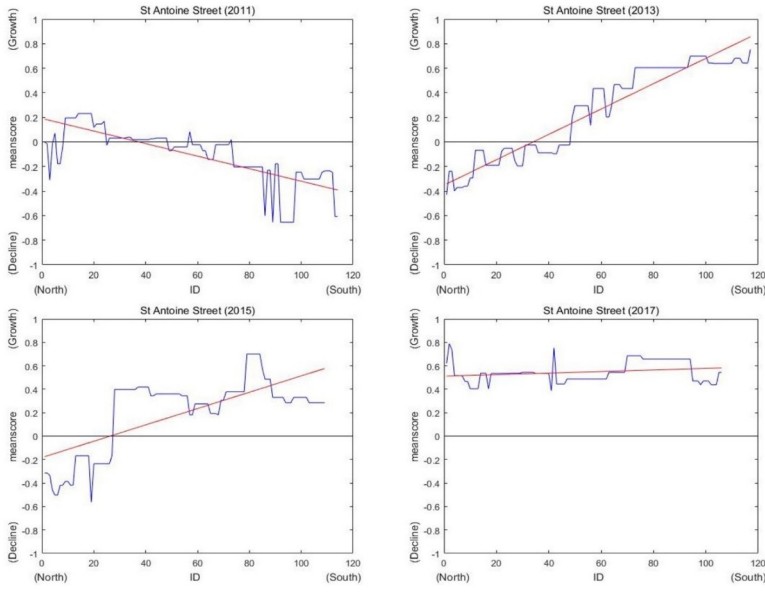

**Fig 14. Graphs of scores on St Antoine Street (2011, 2013, 2015 and 2017).**

Engineering & Environmental Department of the City of Detroit. Each data point shows when the building was permitted by the city. Conversely, crime rate data are a prime indicator of urban decline. Downs [45] used crime rate data to measure the index of urban decline. Crime data are important when attempting to determine whether a city is dangerous or safe. Neighborhood disinvestment, demolition and building activities, demagoguery, and deindustrialization are all factors that have an indirect impact on rising crime rates and fear of crime [47].

## 5.1 Comparison between GSV factor detection and building permits

A building permit is government approval to construct a new building. A building permit indicates that we expect the new construction or extensive refurbishment of a building. A building permit is a sign that a change is happening at the permitted location. We collected building permit data from the Buildings, Safety Engineering & Environmental Department of the City of Detroit, including permit issue dates, permit completion dates, and addresses, and we developed heatmaps of the frequency of building permits in Midtown.

According to the frequency of building permits in 2015 and 2017 in Fig 15, the locations with high scores of urban growth tend to have more building permits. Woodward Avenue and St Antoine Street had numerous permits issued for buildings in 2015 and 2017. By comparing scores and permits between 2015 and 2017, Woodward Avenue shows many changes. High scores were assigned due to the new construction of roads and buildings. The central and southern parts of Woodward Avenue had high scores, and many new building permits occurred there. Additionally, St Antoine Street included a newly permitted building and had a high score.

## 5.2 Comparison between GSV factor detection and crime data

Among the 60 categories of crime, we mainly focus on property-related crimes that include seven categories: larceny, damage to property, burglary, robbery, stolen property, murder, and weapons offenses. The right panels of Fig 16 show heatmaps of crime data from these seven categories for Midtown in 2015 and 2017. High frequencies of crimes are located where the

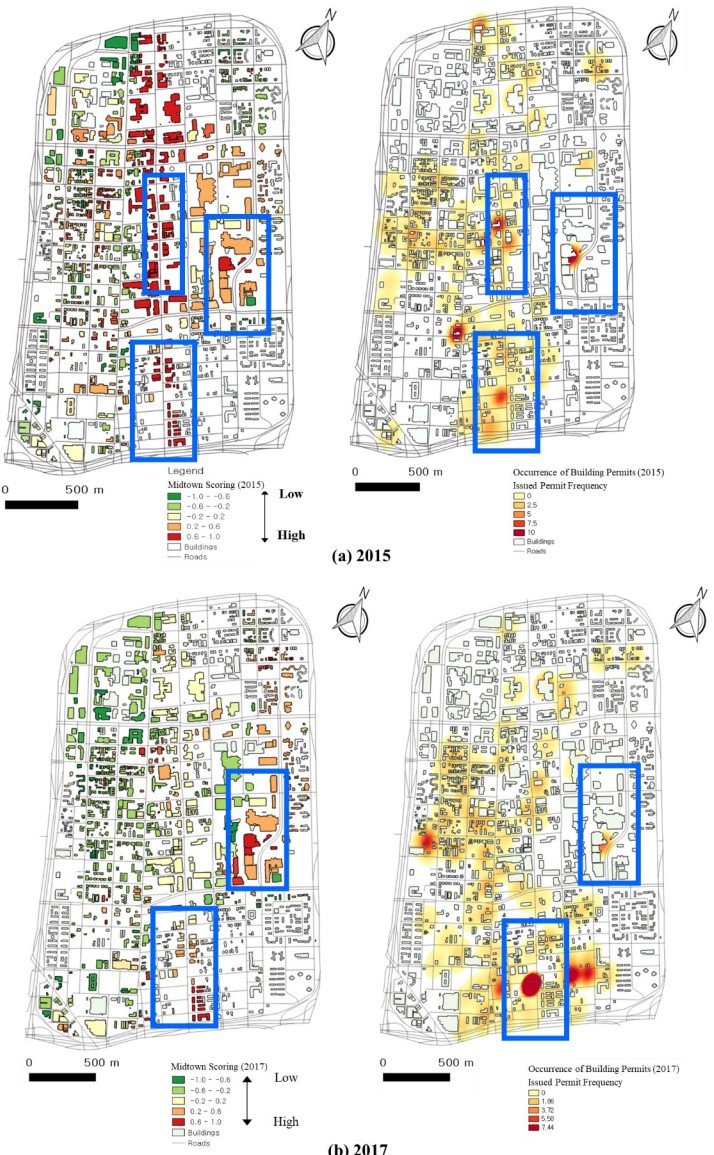

**Fig 15. Scores and building permits in Midtown in (a) 2015 and (b) 2017.**

streets are in decline. High-frequency crime mostly occurs in less changed areas or declining areas, such as Second Avenue and part of Third Avenue.

## 6 Conclusions

This study proposed a new method using computer vision and machine learning to analyze urban growth and decline in an urban area. By training a TensorFlow object detection model with additional objects that affect urban growth and decline, we analyzed changes on streets from Google Street View from 2009 to 2017. By using collected scores, we developed maps of urban growth and decline in Midtown in Detroit, Michigan, USA, as a case study. After determining the scores and statuses for each of the streets, we compared these outcomes with building permit and crime data.

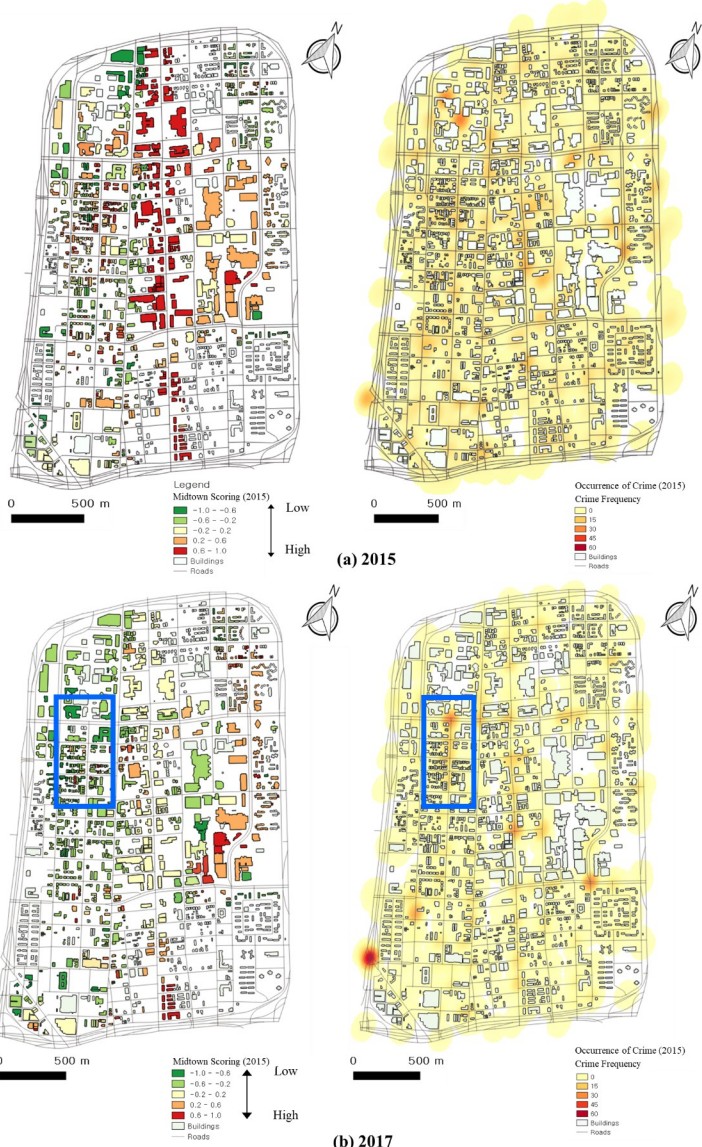

**Fig 16. Scores and crime frequency in Midtown in (a) 2015 and (b) 2017.**

The proposed method allows for the evaluation of urban growth and decline by using street view images. Since Google continuously collects street views, we will be able to understand urban changes and demonstrate urban growth and decline according to the collected data. This study aimed to propose an advanced method of analyzing growth and decline at the street level in urban areas. The advanced approach helps measure urban growth and decline on a specific scope and by visual perception of existing environments. Additionally, the proposed approach is applicable to detect urban changes in other regions. The proposed approach would be useful to examine the status of urban change at a street level both quantitatively and qualitatively. To plan and manage urban areas, the methods and results from this study can be references for developing planning strategies and management schemes in other areas.

This study has limitations. First, scenes from urban streets were taken during particular months of the year. Although we sought to collect data in various locations and years, we

hardly suggest that the scenes represent all day-long situations. Second, the scoring factors were determined by preliminary assumptions. Further studies should be conducted to determine the most appropriate weights for the scores. Although this study illustrates that the proposed method is useful and qualitatively similar to conventional analyses of social and economic data, comparisons using quantitative statistical methods such as regression models would be helpful to validate the detection results of urban growth and decline.

## Author Contributions

**Conceptualization:** Giyoung Byun, Youngchul Kim.

**Data curation:** Giyoung Byun.

**Formal analysis:** Giyoung Byun.

**Funding acquisition:** Youngchul Kim.

**Investigation:** Youngchul Kim.

**Methodology:** Giyoung Byun, Youngchul Kim.

**Project administration:** Youngchul Kim.

**Resources:** Youngchul Kim.

**Software:** Giyoung Byun.

**Supervision:** Youngchul Kim.

**Validation:** Youngchul Kim.

**Visualization:** Giyoung Byun.

**Writing – original draft:** Giyoung Byun.

**Writing – review & editing:** Youngchul Kim.

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
