## [Decision Letter · Decision Letter 0]

21 Oct 2021

PONE-D-21-18677A street-view-based method to detect urban growth and decline: A case study of Midtown in Detroit, Michigan, USAPLOS ONE

Dear Dr. KIM,

Thank you for submitting your manuscript to PLOS ONE. After careful consideration, we feel that it has merit but does not fully meet PLOS ONE’s publication criteria as it currently stands. Therefore, we invite you to submit a revised version of the manuscript that addresses the points raised during the review process.

We look forward to receiving your revised manuscript.

Kind regards,

Wenjia Zhang

Academic Editor

PLOS ONE

Journal Requirements:

“This research was supported by a grant (21TSRD-B151228-03) from Urban Declining Area Regenerative Capacity-Enhancing Technology Research Program and ‘Innovative Talent Education Program for Smart City’ funded by Ministry of Land, Infrastructure and Transport of Korean government and the EDISON Program through the National Research Foundation of Korea (NRF) funded by the Ministry of Science & ICT (NRF-2017M3C1A6075020).”

“The funders had no role in study design, data collection and analysis, decision to publish, or preparation of the manuscript.’

4. We note that Figure 1 & 3 in your submission contain [map/satellite] images which may be copyrighted. All PLOS content is published under the Creative Commons Attribution License (CC BY 4.0), which means that the manuscript, images, and Supporting Information files will be freely available online, and any third party is permitted to access, download, copy, distribute, and use these materials in any way, even commercially, with proper attribution. For these reasons, we cannot publish previously copyrighted maps or satellite images created using proprietary data, such as Google software (Google Maps, Street View, and Earth). For more information, see our copyright guidelines: http://journals.plos.org/plosone/s/licenses-and-copyright.

a. You may seek permission from the original copyright holder of Figures 1 & 3 to publish the content specifically under the CC BY 4.0 license.

We recommend that you contact the original copyright holder with the Content Permission Form (http://journals.plos.org/plosone/s/file?id=7c09/content-permission-form.pdf) and the following text: “I request permission for the open-access journal PLOS ONE to publish XXX under the Creative Commons Attribution License (CCAL) CC BY 4.0 (http://creativecommons.org/licenses/by/4.0/). Please be aware that this license allows unrestricted use and distribution, even commercially, by third parties. Please reply and provide explicit written permission to publish XXX under a CC BY license and complete the attached form.”

b. If you are unable to obtain permission from the original copyright holder to publish these figures under the CC BY 4.0 license or if the copyright holder’s requirements are incompatible with the CC BY 4.0 license, please either i) remove the figure or ii) supply a replacement figure that complies with the CC BY 4.0 license. Please check copyright information on all replacement figures and update the figure caption with source information. If applicable, please specify in the figure caption text when a figure is similar but not identical to the original image and is therefore for illustrative purposes only. The following resources for replacing copyrighted map figures may be helpful:

USGS National Map Viewer (public domain): http://viewer.nationalmap.gov/viewer/ The Gateway to Astronaut Photography of Earth (public domain): http://eol.jsc.nasa.gov/sseop/clickmap/ Maps at the CIA (public domain): https://www.cia.gov/library/publications/the-world-factbook/index.html and https://www.cia.gov/library/publications/cia-maps-publications/index.html NASA Earth Observatory (public domain): http://earthobservatory.nasa.gov/ Landsat: http://landsat.visibleearth.nasa.gov/ USGS EROS (Earth Resources Observatory and Science (EROS) Center) (public domain): http://eros.usgs.gov/# Natural Earth (public domain): http://www.naturalearthdata.com/

Reviewers' comments:

Reviewer's Responses to Questions

**Comments to the Author**

1. Is the manuscript technically sound, and do the data support the conclusions?

Reviewer #1: Yes

Reviewer #2: Yes

2. Has the statistical analysis been performed appropriately and rigorously? 

Reviewer #1: Yes

Reviewer #2: Yes

3. Have the authors made all data underlying the findings in their manuscript fully available?

Reviewer #1: Yes

Reviewer #2: Yes

4. Is the manuscript presented in an intelligible fashion and written in standard English?

Reviewer #1: Yes

Reviewer #2: Yes

5. Review Comments to the Author

Reviewer #1: It is an interesting research providing an interesting view of streets image paste study. The overall writing and the structure should be improved. And there is a lot of redundancy in the writing and presentation of the article please check twice before the next uploading. I have some suggestions as following:

1.Line 26-35 I can understand what you want to say, but please improve the writing and English.

2.Line 121-124 Did you collect the road network of the same area from different years like you did for the street will images?

3.Figure 9, I believe I saw some changes of buildings but why do you use the same base map for all different years? Your messages shows that your research area points on the streets (figure 8 also suggests this) here why do you mark the results on buildings? That’s the buffer methodology you used in Qgis really means this changes are happening in the buildings? I suggest you mark the values on the street directly.

4.Section 5, this part should be a part of results rather than discussion.

Reviewer #2: 1.What are the advantages of identifying urban growth and decline at the street level compared to other methods (estate market conditions, chronological economic status, real incomes, housing conditions)? Could it be proved by relevant literature?

2.Would you describe what gaps are filled by the computational approaches to analyze urban growth and decline used in this paper compared to existing methods?

3.Why did you focus on streets and nearby areas where public transportation routes? Is there literature to support the view?

4.The number of streetscape images collected is inconsistent across years, indicating that the streetscape collection points are inconsistent; are they comparable?

5.Could you indicate the accuracy indicators of deep learning models in the paper?

6.Would you please describe the reasons for the selection of growth and decline factors, and the ratio of the weights assigned?

7.Why is the impact of elements such as bus stops, traffic lights, bicycles rarely described in describing urban change?

8.Why were Woodward Avenue and St Antoine Street are chosen for further discussion and are they representative?

9.Is there any explanation of the significance of comparing GSV characteristics with building permit and crime data?

6. PLOS authors have the option to publish the peer review history of their article (what does this mean?). If published, this will include your full peer review and any attached files.

Reviewer #1: No

Reviewer #2: No

---

## [Author Response · Author response to Decision Letter 0]

30 Nov 2021

Manuscript Number: PONE-D-21-18677

Responses to the reviewers’ comments

“A street-view-based method to detect urban growth and decline: A case study of Midtown in Detroit, Michigan, USA”

Editor

General comments:

Thank you for submitting your manuscript to PLOS ONE. After careful consideration, we feel that it has merit but does not fully meet PLOS ONE’s publication criteria as it currently stands. Therefore, we invite you to submit a revised version of the manuscript that addresses the points raised during the review process.

>> Response & Correction

We would like to thank the editor for encouraging us to revise the manuscript. We strongly believe that in the revision we have fully addressed all comments and carefully revised the manuscript accordingly to the comments that we received.

Please see the following responses to the comments. Thank you very much.

[E-1] 

1. We note that Figure 1 & 3 in your submission contain [map/satellite] images which may be copyrighted. All PLOS content is published under the Creative Commons Attribution License (CC BY 4.0), which means that the manuscript, images, and Supporting Information files will be freely available online, and any third party is permitted to access, download, copy, distribute, and use these materials in any way, even commercially, with proper attribution. For these reasons, we cannot publish previously copyrighted maps or satellite images created using proprietary data, such as Google software (Google Maps, Street View, and Earth). For more information, see our copyright guidelines: http://journals.plos.org/plosone/s/licenses-and-copyright. 

a. You may seek permission from the original copyright holder of Figures 1 & 3 to publish the content specifically under the CC BY 4.0 license. 

We recommend that you contact the original copyright holder with the Content Permission Form (http://journals.plos.org/plosone/s/file?id=7c09/content-permission-form.pdf) and the following text: “I request permission for the open-access journal PLOS ONE to publish XXX under the Creative Commons Attribution License (CCAL) CC BY 4.0 (http://creativecommons.org/licenses/by/4.0/). Please be aware that this license allows unrestricted use and distribution, even commercially, by third parties. Please reply and provide explicit written permission to publish XXX under a CC BY license and complete the attached form.” 

b. If you are unable to obtain permission from the original copyright holder to publish these figures under the CC BY 4.0 license or if the copyright holder’s requirements are incompatible with the CC BY 4.0 license, please either i) remove the figure or ii) supply a replacement figure that complies with the CC BY 4.0 license. Please check copyright information on all replacement figures and update the figure caption with source information. If applicable, please specify in the figure caption text when a figure is similar but not identical to the original image and is therefore for illustrative purposes only. The following resources for replacing copyrighted map figures may be helpful: 

USGS National Map Viewer (public domain): http://viewer.nationalmap.gov/viewer/ The Gateway to Astronaut Photography of Earth (public domain): http://eol.jsc.nasa.gov/sseop/clickmap/ Maps at the CIA (public domain): https://www.cia.gov/library/publications/the-world-factbook/index.html and https://www.cia.gov/library/publications/cia-maps-publications/index.html NASA Earth Observatory (public domain): http://earthobservatory.nasa.gov/ Landsat: http://landsat.visibleearth.nasa.gov/ USGS EROS (Earth Resources Observatory and Science (EROS) Center) (public domain): http://eros.usgs.gov/# Natural Earth (public domain): http://www.naturalearthdata.com/. 

>> Response & Correction

Thank you for your comments and suggestion. Accordingly, we have revised a base map in Fig 1 from USGS National Map Viewer (public domain: http://viewer.nationalmap.gov/viewer). Since we have deleted a street view image in Fig 3, Fig 3 does not include a map or satellite image. 

[E-2] 

>> Response & Correction

Thank you for the comments. I have double-checked the grant information in the Funding Information and Financial Disclosure. Now I believe all information are correct. .

Reviewer 1

General comments:

It is an interesting research providing an interesting view of streets image paste study. The overall writing and the structure should be improved. And there is a lot of redundancy in the writing and presentation of the article please check twice before the next uploading. I have some suggestions as following: 

>> Response & Correction

We would like to thank the reviewer 1 for encouraging us to revise the manuscript. According to your comments and suggestion, we have carefully revised the manuscript. Please see the following responses to the comments. Thank you very much.

[R1-1] 

1. Line 26-35 I can understand what you want to say, but please improve the writing and English. 

>> Response & Correction

Thank you for your comment and suggestion. As you mentioned, we have sought to explain the urban growth and decline more clearly. According to your comment and suggestion, we have carefully improved the manuscript. Additionally, we have thoroughly edited grammatical errors and use of language with professional editors. In particular, we have revised the paragraph in lines 26 to 35. It reads:

“Urban growth and decline are processes that show changes in urban areas. While urban growth is a process to newly create various urban elements [1], urban decline means that the entire or part of an urban area becomes devastated and abandoned. For example, urban growth includes spatial extension, population increase, and economic growth [2,3], and urban decline includes empty houses, abandoned buildings, and job losses [3,4]. Since urban decline is an undesirable condition in cities, various policies are used to try to change and protect urban areas from urban decline. When analyzing the situation of urban growth and decline, it is possible to prepare for urban planning and management to sustain daily places in urban areas. Thus, as urban growth and decline have occurred, relevant studies have examined and proposed models to analyze urban growth and decline.”

[R1-2] 

2. Line 121-124 Did you collect the road network of the same area from different years like you did for the street will images? 

>> Response & Correction

Unfortunately, we could not collect the road networks in 2011, 2013, 2015 and 2017. However since the road network in the area rarely changed during those years, we decided to use existing conditions in 2018 to compare with other years. So, we used OpenStreetMap API data of buildings, blocks, and streets in December 2018. To make it clear, we have added sentences in lines 127 to 128 as below.

“… We used OpenStreetMap Data including buildings, blocks, and streets in December 2018 as a base map to compare. ….”

[R1-3] 

3. Figure 9, I believe I saw some changes of buildings but why do you use the same base map for all different years? Your messages shows that your research area points on the streets (figure 8 also suggests this) here why do you mark the results on buildings? That’s the buffer methodology you used in Qgis really means this changes are happening in the buildings? I suggest you mark the values on the street directly. 

>> Response & Correction

Thank you for your comment and suggestion. Our main purpose was to measure urban growth and decline at the street level with a deep learning method using street image data. To achieve this research purpose, we sought to visualize the urban growth and decline scores on the map. Since we collected the street-view images at two-year intervals, the number of images and locations of viewing points are different between years. Thus, we used a buffer of 200 m to supplement these differences. Because the spots of captured street-view images of each year are a little different and inconsistent, we found buildings intersecting with a buffer of 200 m and calculated the mean of score change in those buildings. In particular, since we could not collect relevant maps in different years, we decided to use the map of December 2018 as base map to compare. To make it clearer, we have added additional explanation in lines 287 to 293. It reads: 

“To visualize the urban growth and decline analysis, we create maps using QGIS. Because the scoring is based on panoramic street views and the captured points of street view images each year are different, we determined that the best way to score growth and decline is to express the score using a buffer of 200 meters for the locations of the street view images. Fig. 8 shows the locations of the street view images in 2017. By using an intersection function in QGIS, we combine the scores with the building data within the buffer. By extracting the mean values of urban street scores, we determine whether an urban street is growing or declining. ….”

[R1-4] 

4. Section 5, this part should be a part of results rather than discussion. 

>> Response & Correction

Thank you for your comment. The reason that we put this part in the section of Discussion is to compare between the results by our proposed vision method and existing statistical data. As we mentioned in the section of Introduction, previous studies about analyzing urban growth and decline usually focused on social and economic aspects within a wide-range whole city. To supplement these aspects, our research has focused on a computational method with a specific urban area. Thus, we have verified our results with social and economic aspects. To make it clearer, we have revised the section of Discussion in lines 380 to 393 as below: 

“To verify the results of growth and decline in Midtown, we compare the results of growth and decline with building permits and crime data. A building permit is a type of permission granted by a government or other regulatory agency for the lawful construction of new buildings and extensive work of existing buildings [44]. When any form of work that will modify or add structure to an existing property or land parcel is planned, building permits are necessary [45]. In terms of urban planning, building permits are a necessary part of well-planned urban development. Tasantab [46] used building permits to see how they may be utilized as effective controls on physical development. Building permit data are issued by the Buildings, Safety Engineering & Environmental Department of the City of Detroit. Each data point shows when the building was permitted by the city. Conversely, crime rate data are a prime indicator of urban decline. Downs [47] used crime rate data to measure the index of urban decline. Crime data are important when attempting to determine whether a city is dangerous or safe. Neighborhood disinvestment, demolition and building activities, demagoguery, and deindustrialization are all factors that have an indirect impact on rising crime rates and fear of crime [48]. …”

Reviewer 2

[R2-1] 

1. What are the advantages of identifying urban growth and decline at the street level compared to other methods (estate market conditions, chronological economic status, real incomes, housing conditions)? Could it be proved by relevant literature? 

>> Response & Correction

Thank you for your comments. Unlike other methods, our proposed method has a strength to identify urban changes at the street level. Identifying urban changes at the street level helps develop more specific planning strategies focusing on streets where people daily explore and easily identify the improvement. Accordingly, we have carefully added relevant literature reviews and explanation in the section of Introduction in lines 38 to 48 as below:

“… Those studies demonstrated theoretical and empirical evidence to understand urban growth and decline. Using spatial and statistical data, an urban growth model was proposed to include commonly available, spatially specific data points [10]. Those previous studies have a strength to analyze and apply for every region and country using spatial and statistical data. Although those relevant studies effectively explained the characteristics of urban growth and decline, they normally demonstrate overall changes and rarely explain street-level changes where people explore for their daily life. Since people explore places and notice changes daily at the street level, it would be useful to develop a method to identify urban changes at the street level and demonstrate whether urban growth or decline occurs there. Street-level analysis helps develop local-oriented specific strategies focusing on where people daily explore and easily identify changes for the region by appropriate solutions.…”

[R2-2] 

2. Would you describe what gaps are filled by the computational approaches to analyze urban growth and decline used in this paper compared to existing methods? 

>> Response & Correction

Thank you for your comments. According to your comment, we have added the strength of existing method and our approaches in the section of Introduction in lines 41 to 45. Additionally, we have revised the conclusion including what gaps are filled compared to existing methods in lines 426 to 430. We strongly believe that it would be helpful for better understanding what gaps we have filled between computational methods and existing methods.

Line 41-45

“… Those previous studies have a strength to analyze and apply for every region and country using spatial and statistical data. Although those relevant studies effectively explained the characteristics of urban growth and decline, they normally demonstrate overall changes and rarely explain street-level changes where people explore for their daily life.. …”

Line 434-439

“… This study aimed to propose an advanced method of analyzing growth and decline at the street level in urban areas. The advanced approach helps measure urban growth and decline on a specific scope and by visual perception of existing environments. Additionally, the proposed approach is applicable to detect urban changes in other regions. The proposed approach would be useful to examine the status of urban change at a street level both quantitatively and qualitatively. …”

[R2-3] 

3. Why did you focus on streets and nearby areas where public transportation routes? Is there literature to support the view?

>> Response & Correction

We empirically assumed that public transportation roads are routes where people and vehicles move frequently. According to your comment, we have cited relevant references about transportation routes and urban changes in lines 129 to 132. It reads:

“… Various daily activities, such as commuting and shopping, normally occur along these public transportation routes. Public transportation routes are an important factor when planning sustainable cities and urban development [26]. Additionally, newly urban areas are developed along major transportation routes [27]. …”

[R2-4] 

4. The number of streetscape images collected is inconsistent across years, indicating that the streetscape collection points are inconsistent; are they comparable?

>> Response & Correction

Thank you for your comment and suggestion. Our main purpose was to measure urban growth and decline at the street level with a deep learning method using street image data. To achieve this research purpose, we sought to visualize the urban growth and decline scores on the map. Since we collected the street-view images at two-year intervals, the number of images and locations of viewing points are different between years. Thus, we used a buffer of 200 m to supplement these differences. Because the spots of captured street-view images of each year are a little different and inconsistent, we found buildings intersecting with a buffer of 200 m and calculated the mean of score change in those buildings. In particular, since we could not collect relevant maps in different years, we decided to use the map of December 2018 as base map to compare. To make it clearer, we have added additional explanation in lines 287 to 293. It reads: 

“To visualize the urban growth and decline analysis, we create maps using QGIS. Because the scoring is based on panoramic street views and the captured points of street view images each year are different, we determined that the best way to score growth and decline is to express the score using a buffer of 200 meters for the locations of the street view images. Fig. 8 shows the locations of the street view images in 2017. By using an intersection function in QGIS, we combine the scores with the building data within the buffer. By extracting the mean values of urban street scores, we determine whether an urban street is growing or declining. ….”

[R2-5] 

5. Could you indicate the accuracy indicators of deep learning models in the paper?

>> Response & Correction

Thank you for your comments and suggestion. Accordingly, we have carefully reviewed your suggestion and included additional explanation about the accuracy indicators of our models in lines 208 to 210 as below.

“… The accuracy indicator of the deep learning models we trained was approximately 86%, and examples of detected results for each element are shown in Figs 4 to 7.”

[R2-6] 

6. Would you please describe the reasons for the selection of growth and decline factors, and the ratio of the weights assigned?

>> Response & Correction

Thank you for your comments. To choose growth and decline factors, we investigated elements that we can identify in collected street view images. To determine the ratio of the weights, we reviewed suggestions in relevant references and considered its impacts on cognition by its size and frequency. To make it clearer, we have added relevant sentences in Section 3.3.2 in lines 230 to 265 as below:

“We chose six growth factors to evaluate the growth score and its impacts on the spatial change by considering its size and frequency in collected street view images. The first factor is buildings. If a new building is in an image compared to images from past years, it means that the street has been developed [28]. The more buildings are constructed, the greater the population density. As the number of buildings increases, urbanization and urban growth are shown [29]. Thus, we score +0.5 for new buildings. Additionally, the new construction of buildings can be a source of economic capacity or can increase economic capacity [30]. The second factor is a crane, which indicates a construction site. It is a main factor to construction site. A crane should inevitably be used when a building is constructed [31]. When a crane is captured, we score +0.3 for that image. The third factor is a traffic barrel, which is used to signify a construction site. Traffic barrels are a common sign that road work or other types of construction take place near or on a road [32]. If a street has a construction site and traffic barrels, we score +0.03 for each barrel, showing that urban growth is in the construction process. Constructing new buildings and housing is a significant factor that improves the economic conditions in a city [33]. The fourth factor is a bus stop. If we detect new bus stops, we score them +0.2 points each. Bus stops and traffic lights are parts of transportation systems. Accessibility to bus stops is usually used to measure urban growth and can be an indicator of urban sprawl [34,35]. Since changes in urban transportation can affect urban employment [36], new components of transportation change public space. For the fifth factor, if there is a new traffic light, we score +0.1 points for urban growth. Traffic Lights show the traffic volume of vehicles [37]. They can be urbanization infrastructures as traffic management systems [38]. The sixth factor includes people, cars, and bicycles, which can affect urban activation and traffic flow. For street activation, walkways and bikeways are necessary [39]. If the traffic flow of the region is concentrated, infrastructural facilities are located at the center of the town [40]. Scores of people, cars, and bicycles are given +0.02 points each.

Regarding the decline factors on a street, we chose the loss of similar factors. The first factor is the loss of a building. The loss of a building has a huge impact on a street. Although the loss of a building can be explained in many ways, deconstruction obviously occurs at the location. Since deconstruction can affect the deactivation of a street [41], we score the loss of a building at -0.5 points. For the second decline factor, if there is no person in the view, we score this situation at -0.02 points each. We easily observe many people on an active street. People go shopping or walking on a street if the street is alive. For the third decline factor, if no cars are detected in the view, we score such scenes at -0.02 points each. The disappearance of the growth factors, crane and traffic barrel would mean the end of the construction, so we excluded those factors. Table 2 shows the growth and decline factors used in this study. By detecting these factors on urban streets, we calculated urban growth and decline scores. ”

[R2-7] 

7. Why is the impact of elements such as bus stops, traffic lights, bicycles rarely described in describing urban change?

>> Response & Correction

Thank you for your comments and suggestion. To make it clearer, we have revised the Table 2 and added additional explanation in lines 230-266 as below: 

“We chose six growth factors to evaluate the growth score and its impacts on the spatial change by considering its size and frequency in collected street view images. The first factor is buildings. If a new building is in an image compared to images from past years, it means that the street has been developed [28]. The more buildings are constructed, the greater the population density. As the number of buildings increases, urbanization and urban growth are shown [29]. Thus, we score +0.5 for new buildings. Additionally, the new construction of buildings can be a source of economic capacity or can increase economic capacity [30]. The second factor is a crane, which indicates a construction site. It is a main factor to construction site. A crane should inevitably be used when a building is constructed [31]. When a crane is captured, we score +0.3 for that image. The third factor is a traffic barrel, which is used to signify a construction site. Traffic barrels are a common sign that road work or other types of construction take place near or on a road [32]. If a street has a construction site and traffic barrels, we score +0.03 for each barrel, showing that urban growth is in the construction process. Constructing new buildings and housing is a significant factor that improves the economic conditions in a city [33]. The fourth factor is a bus stop. If we detect new bus stops, we score them +0.2 points each. Bus stops and traffic lights are parts of transportation systems. Accessibility to bus stops is usually used to measure urban growth and can be an indicator of urban sprawl [34,35]. Since changes in urban transportation can affect urban employment [36], new components of transportation change public space. For the fifth factor, if there is a new traffic light, we score +0.1 points for urban growth. Traffic Lights show the traffic volume of vehicles [37]. They can be urbanization infrastructures as traffic management systems [38]. The sixth factor includes people, cars, and bicycles, which can affect urban activation and traffic flow. For street activation, walkways and bikeways are necessary [39]. If the traffic flow of the region is concentrated, infrastructural facilities are located at the center of the town [40]. Scores of people, cars, and bicycles are given +0.02 points each.

Regarding the decline factors on a street, we chose the loss of similar factors. The first factor is the loss of a building. The loss of a building has a huge impact on a street. Although the loss of a building can be explained in many ways, deconstruction obviously occurs at the location. Since deconstruction can affect the deactivation of a street [41], we score the loss of a building at -0.5 points. For the second decline factor, if there is no person in the view, we score this situation at -0.02 points each. We easily observe many people on an active street. People go shopping or walking on a street if the street is alive. For the third decline factor, if no cars are detected in the view, we score such scenes at -0.02 points each. The disappearance of the growth factors, crane and traffic barrel would mean the end of the construction, so we excluded those factors. Table 2 shows the growth and decline factors used in this study. By detecting these factors on urban streets, we calculated urban growth and decline scores. ”

[R2-7] 

8. Why were Woodward Avenue and St Antoine Street are chosen for further discussion and are they representative?

>> Response & Correction

According to the results, two streets demonstrate the significant changes with growth and decline score. The Woodward Avenue is the main street of Midtown, Detroit. It is located in the center of Midtown, and also have lots of facilities and amenities. St Antoine Street contains various medical centers where people go to receive treatments and get vaccinated including children hospital. To make it clearer, we have added some explanation in lines 337-342 as below:

“... . After investigating the mean scores of all the streets in Midtown, Woodward Avenue and St Antoine Street are chosen for further discussion of the graphs of their growth and decline scores. Woodward Avenue is a main road that has a long history and important facilities [42]. Additionally, St Antoine Street is a representative area for medical services [43]. …”

[R2-8] 

9. Is there any explanation of the significance of comparing GSV characteristics with building permit and crime data?

>> Response & Correction

Thank you for your comments and suggestion. We have sought to verify the vision-based result of urban growth and decline with social and economic data. Accordingly, we used building permit and crime data as a representative verification process. To make it clearer, we have added some literature review in Section 5 in Line 377-390 as below: 

“To verify the results of growth and decline in Midtown, we compare the results of growth and decline with building permits and crime data. A building permit is a type of permission granted by a government or other regulatory agency for the lawful construction of new buildings and extensive work of existing buildings [44]. When any form of work that will modify or add structure to an existing property or land parcel is planned, building permits are necessary [45]. In terms of urban planning, building permits are a necessary part of well-planned urban development. Tasantab [46] used building permits to see how they may be utilized as effective controls on physical development. Building permit data are issued by the Buildings, Safety Engineering & Environmental Department of the City of Detroit. Each data point shows when the building was permitted by the city. Conversely, crime rate data are a prime indicator of urban decline. Downs [47] used crime rate data to measure the index of urban decline. Crime data are important when attempting to determine whether a city is dangerous or safe. Neighborhood disinvestment, demolition and building activities, demagoguery, and deindustrialization are all factors that have an indirect impact on rising crime rates and fear of crime [48]. …”

---

## [Decision Letter · Decision Letter 1]

5 Jan 2022

PONE-D-21-18677R1A street-view-based method to detect urban growth and decline: A case study of Midtown in Detroit, Michigan, USAPLOS ONE

Dear Dr. KIM,

Thank you for submitting your manuscript to PLOS ONE. After careful consideration, we feel that it has merit but does not fully meet PLOS ONE’s publication criteria as it currently stands. Therefore, we invite you to submit a revised version of the manuscript that addresses the points raised during the review process.

We look forward to receiving your revised manuscript.

Kind regards,

Wenjia Zhang

Academic Editor

PLOS ONE

Journal Requirements:

Reviewers' comments:

Reviewer's Responses to Questions

**Comments to the Author**

1. If the authors have adequately addressed your comments raised in a previous round of review and you feel that this manuscript is now acceptable for publication, you may indicate that here to bypass the “Comments to the Author” section, enter your conflict of interest statement in the “Confidential to Editor” section, and submit your "Accept" recommendation.

Reviewer #1: All comments have been addressed

Reviewer #2: (No Response)

2. Is the manuscript technically sound, and do the data support the conclusions?

Reviewer #1: Yes

Reviewer #2: Yes

3. Has the statistical analysis been performed appropriately and rigorously? 

Reviewer #1: Yes

Reviewer #2: Yes

4. Have the authors made all data underlying the findings in their manuscript fully available?

Reviewer #1: Yes

Reviewer #2: Yes

5. Is the manuscript presented in an intelligible fashion and written in standard English?

Reviewer #1: Yes

Reviewer #2: Yes

6. Review Comments to the Author

Reviewer #1: The author has addressed all the comments I made in the previous version. And I think this paper is qualified enough to be published.

Reviewer #2: By the last round of revision, the authors have addressed my concerns. Then I would like to recommend the editor broad to accept this paper. Before publishing, the author should improve again the writing quality of this manuscript, and especially focus on the introduction part and state more clearly about the research motivation and contributions.

7. PLOS authors have the option to publish the peer review history of their article (what does this mean?). If published, this will include your full peer review and any attached files.

Reviewer #1: No

Reviewer #2: No

---

## [Author Response · Author response to Decision Letter 1]

11 Jan 2022

Manuscript Number: PONE-D-21-18677

Responses to the reviewers’ comments

“A street-view-based method to detect urban growth and decline: A case study of Midtown in Detroit, Michigan, USA”

Reviewer 1

General comments:

Reviewer #1: The author has addressed all the comments I made in the previous version. And I think this paper is qualified enough to be published.

>> Response & Correction

We would like to thank the reviewer 1 for reviewing our manuscript and giving us valuable suggestion and comments. With your comments and suggestion, we improved our manuscript. Thank you very much.

Reviewer 2

General comments:

Reviewer #2: By the last round of revision, the authors have addressed my concerns. Then I would like to recommend the editor broad to accept this paper. Before publishing, the author should improve again the writing quality of this manuscript, and especially focus on the introduction part and state more clearly about the research motivation and contributions.

>> Response & Correction

We would like to thank the reviewer 2 for supporting our revision and suggesting the improvement. According to your suggestion, we have carefully revised the manuscript. In particular, we have added motivation and contribution in the section of Introduction. It reads: 

“… In particular, yearly changes in street view images motivate this study to focus on urban changes at a street level. This novel method to analyze urban changes at a street level seeks to contribute to automation and efficiency in identifying growth and declines in urban areas. ….”

---

## [Editor Report · Decision Letter 2]

27 Jan 2022

A street-view-based method to detect urban growth and decline: A case study of Midtown in Detroit, Michigan, USA

PONE-D-21-18677R2

Dear Dr. KIM,

We’re pleased to inform you that your manuscript has been judged scientifically suitable for publication and will be formally accepted for publication once it meets all outstanding technical requirements.

Kind regards,

Wenjia Zhang

Academic Editor

PLOS ONE
---

## [Editor Report · Acceptance letter]

30 Jan 2022

PONE-D-21-18677R2 

A street-view-based method to detect urban growth and decline: A case study of Midtown in Detroit, Michigan, USA 

Dear Dr. KIM:

I'm pleased to inform you that your manuscript has been deemed suitable for publication in PLOS ONE. Congratulations! Your manuscript is now with our production department. 

Kind regards, 

on behalf of

Dr. Wenjia Zhang 

Academic Editor

PLOS ONE